# UniTime: A Language-Empowered Unified Model for Cross-Domain Time Series Forecasting

## ABSTRACT

Multivariate time series forecasting plays a pivotal role in contemporary web technologies. In contrast to conventional methods that involve creating dedicated models for specific time series application domains, this research advocates for a unified model paradigm that transcends domain boundaries. However, learning an effective cross-domain model presents the following challenges. First, various domains exhibit disparities in data characteristics, e.g., the number of variables, posing hurdles for existing models that impose inflexible constraints on these factors. Second, the model may encounter difficulties in distinguishing data from various domains, leading to suboptimal performance in our assessments. Third, the diverse convergence rates of time series domains can also result in compromised empirical performance. To address these issues, we propose UniTime for effective cross-domain time series learning. Concretely, UniTime can flexibly adapt to data with varying characteristics. It also uses domain instructions and a Language-TS Transformer to offer identification information and align two modalities. In addition, UniTime employs masking to alleviate domain convergence speed imbalance issues. Our extensive experiments demonstrate the effectiveness of UniTime in advancing state-of-the-art forecasting performance and zero-shot transferability.

## ACM Reference Format:

Anonymous Author(s). 2018. UniTime: A Language-Empowered Unified Model for Cross-Domain Time Series Forecasting. In *Proceedings of Make sure to enter the correct conference title from your rights confirmation emai (Conference acronym 'XX)*. ACM, New York, NY, USA, 11 pages. https://doi.org/XXXXXXX.XXXXXXX

## 1 INTRODUCTION

The World Wide Web, as a dynamic and ever-evolving ecosystem, relies heavily on the ability to anticipate and adapt to changing patterns and user behaviors. Multivariate time series forecasting, with its capacity to analyze historical data and predict future trends, emerges as a crucial tool in modern web technologies [8, 10, 11, 13, 35]. The capability of accurate forecasts has the potential not only to enhance user experiences but also to drive the development of intelligent web services, such as content recommendations [30], web economics modeling [35], microservice logs analysis [12], as well as early warning systems against emerging threats [11].

Recently, Transformers [28] have achieved exceptional performance in various tasks of natural language processing [14, 25, 26] and computer vision [3, 5, 21], which also triggered significant interest in the time series community [31]. Benefiting from the self-attention mechanism to capture long-range temporal dependencies in sequential data, a multitude of Transformer-based models have been proposed for time series forecasting [16, 19, 20, 24, 32, 34, 40, 41]. This rapid progress has consistently pushed the boundaries of state-of-the-art performance in forecasting benchmarks from diverse application domains, including energy, economics, weather, transportation, and disease predictions.

While these models have shown impressive performance, they employ a strategy of training a dedicated model for each domain (or dataset). We argue that this approach may be overly restrictive and overlooks the potential benefits of training a unified model capable of generalizing across various domains. Such a unified model paradigm has achieved remarkable success in computer vision [15, 23], natural language processing [1, 25], and holds promise in the context of time series modeling. An illustration of the two paradigms are presented in Figure 1.

The advantage of training a cross-domain time series model lies in its ability to leverage abundant data from diverse domains with varying temporal characteristics. This enables the model to learn the underlying commonalities present in time series data, which are intrinsic and shared across domains. For instance, while the specific patterns of seasonality (e.g., daily or weekly) may differ between domains, the fundamental concept of recurring patterns within the data is a shared characteristic. Additionally, the presence of trends (e.g., upward or downward) may vary from one domain to another, but the shared property is the recognition of data evolving over time. By equipping the model with this generalization capability, it stands to benefit from enhanced predictive performance and the ability to transfer its knowledge to previously unseen domains. This potential for broader applicability, improved performance, and streamlined deployment underscores the value of cross-domain time series modeling. However, to effectively learn a unified model for data from diverse domains is technically non-trivial, with the following three challenges.

- **Varying Data Characteristics.** Data from various domains exhibit differences in the number of variables (channels), lengths of histories, and lengths of future predictions. However, existing model designs typically impose rigid constraints on these factors, limiting their ability to generalize across domains. For instance, many approaches employ the channel mixing design [19, 32, 33], which locks the number of input channels to a constant value, making it nearly infeasible to implement a shared encoder capable of handling time series from domains with distinct semantics.
- **Domain Confusion Issue.** When training a model across multiple time series domains, especially when these domains display notable variations in temporal patterns or distributions [20, 34],

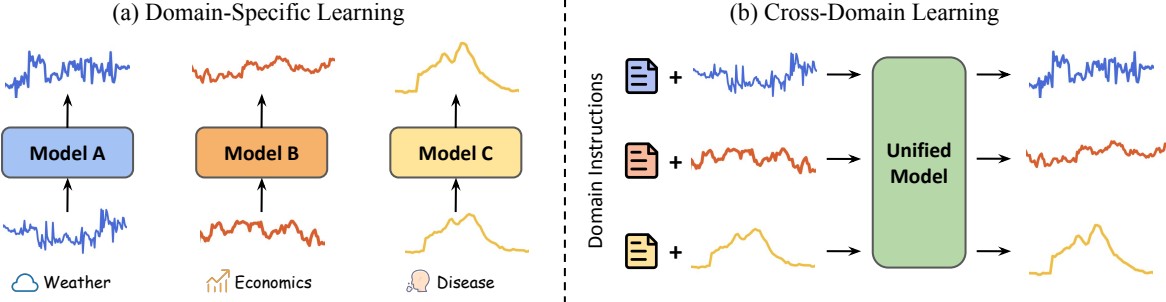

(a) Domain-Specific Learning

(b) Cross-Domain Learning

**Figure 1: (a) Specialized models are separately trained on time series domains with notable distribution differences. For instance, weather time series constantly fluctuate due to the chaotic influence of natural factors, while economic data, such as exchange rates, tends to remain relatively stable. Disease data, like seasonal cold patterns, typically demonstrate periodicity over extended time periods. (b) The proposed cross-domain learning approach handles time series data from distinct domains and utilizes natural language as domain instructions to provide domain-specific information.**

the model may struggle with discerning and adapting to these differences. This challenge, termed *domain confusion* in this study, results in subpar empirical performance.

- **Domain Convergence Speed Imbalance.** Various time series domains exhibit diverse convergence rates attributed to their unique characteristics. For instance, domains with simple and regular patterns may rapidly reach convergence during model training, and then exhibit a tendency for overfitting, whereas others may require more iterations to achieve convergence. Experimentally, this disparity in learning dynamics leads to a compromise in cross-domain forecasting performance.

To address the aforementioned challenges, this paper introduces **UniTime**, an innovative solution for effectively learning from cross-domain time series data. First, UniTime offers flexibility in its overall design, accommodating time series data with varying characteristics, e.g., input and output lengths. Second, inspired by the recent progress in language instruction-based model tuning [4, 29, 38, 39], we propose the use of human-crafted instructions to furnish the model with explicit domain identification information, alleviating the issue of domain confusion. We further introduce a Language-TS Transformer designed to process both instructions and time series. Thus, time series from different input spaces are aligned to the common latent space of language models, facilitating cross-domain generalization. Third, we employ masking to mitigate the problem of domain convergence speed imbalance, by constraining the model from acquiring trivial solutions, such as memorizing exclusive data patterns, on domains susceptible to overfitting. Our contributions are summarized below.

- To the best of our knowledge, we present the first attempt to explore the potential of using a unified model for generalization across time series application domains.
- We propose UniTime as a versatile model, which is capable of handling time series data with varying characteristics, distinguishing between different domains, and balancing data with diverse convergence rates.
- Our extensive experiments affirm the effectiveness of UniTime. It attains new state-of-the-art performance on popular time series forecasting benchmarks, and showcases admirable transferability to unseen domains.

## 2 RELATED WORK

**Deep Models for Time Series Forecasting.** Deep learning models with elaborately crafted architectures have demonstrated great promise in time series forecasting. Among them, Transformer-based models have gained widespread recognition due to their exceptional prowess in sequence modeling [31]. However, the self-attention mechanisms in Transformers are known to introduce high computational and memory complexities. Consequently, a plethora of approaches, such as LogTrans [18], Reformer [16], Informer [40], Pyraformer [19] have been proposed to reduce the cost for better efficiency. Another line of research concentrates on capturing the intricate temporal patterns within time series data by leveraging techniques such as seasonal-trend decomposition (Autoformer [34], ETSformer [32], FEDformer [41]) and non-stationary information compensation (NSformer [20]), so as to boost performance. Recently, the community has initiated efforts to develop more versatile methods. For example, TimesNet [33] proposes a generic framework to tackle multiple time series tasks. Following TimesNet, GPT4TS [42] proposes to leverage pretrained language models to process time series signals. However, the above methods still employ separate models for each domain/dataset, limiting their potential to become the foundational architecture for general time series modeling.

**Language Model Powered Cross-Modality Learning.** Recently, there has been a notable surge of interest in the utilization of pretrained language models to other research fields with distinct modalities, including recommendation systems [9, 38], graph learning [7, 36, 39], and time series modeling [42]. For instance, InstructRec [38] reformulates recommendation tasks into text form, utilizing instructions to enable language models to generate recommendations. GIMLET [39] employs natural language to describe tasks, which not only allows the incorporation of textual knowledge, but also empowers models to accomplish molecule-related tasks using specific instructions. GPT4TS [42] is a relevant work to this study, as it also employs language models to forecast the future. While GPT4TS demonstrates the feasibility of processing time series with language models, it primarily relies on a single modality, namely the time series data itself. It falls short of fully exploiting the powerful language processing capabilities that language models offer, which are pivotal in facilitating cross-domain time series learning.

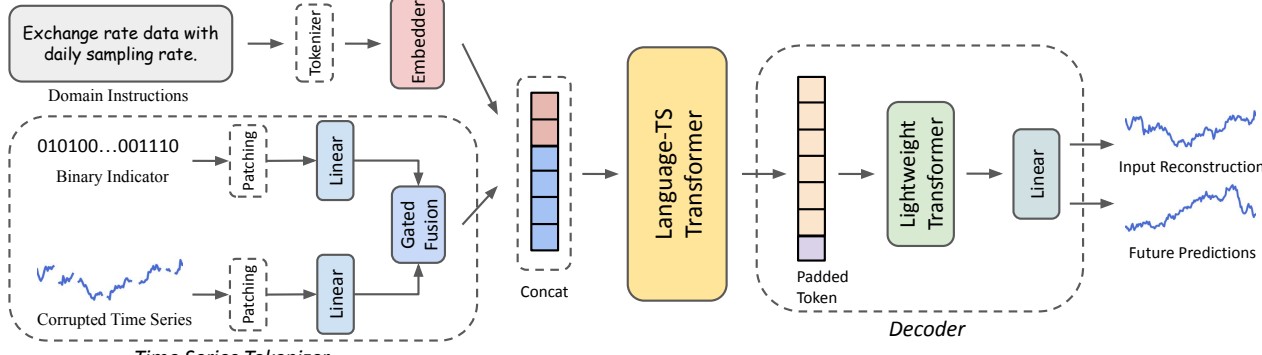

**Figure 2: UniTime overview from the perspective of a univariate time series.**

## 3 PRELIMINARIES

**Problem Definition.** The primary emphasis of this study lies in the development of cross-domain time series models. To this end, we define an observation of a multivariate time series from domain $\tau$ at time step $t$ as $\boldsymbol{x}_\tau^t = \{x_{\tau,1}^t, ..., x_{\tau,c_\tau}^t\} \in \mathbb{R}^{c_\tau}$, where $c_\tau$ represents the number of channels or variables within domain $\tau$. In the context of cross-domain time series forecasting, both the historical and future prediction lengths can vary across domains. Thus, we use $L_\tau$ to denote the lookback window and $T_\tau$ to denote the future prediction range in domain $\tau$, and represent the input and output of the model as $X_\tau^{L_\tau} = \{\boldsymbol{x}_\tau^1, ..., \boldsymbol{x}_\tau^{L_\tau}\} \in \mathbb{R}^{L_\tau \times c_\tau}$ and $\hat{X}_\tau^{T_\tau} = \{\hat{\boldsymbol{x}}_\tau^{L_\tau+1}, ..., \hat{\boldsymbol{x}}_\tau^{L_\tau+T_\tau}\} \in \mathbb{R}^{T_\tau \times c_\tau}$.

**Channel-Mixing v.s. Channel-Independence.** Many time series Transformer models typically adopt a channel-mixing configuration [33, 34, 40, 41]. In this setup, an embedding layer is utilized to process data from all time series channels and project them into a hidden space for multi-channel information fusion. However, this setting poses challenges when attempting to train models across time series domains due to two key issues: (1) the number of channels typically varies among different time series domains, and (2) employing a shared embedding layer to process time series channels from different domains with significantly distinct semantics is impractical. To tackle the problems, our study embraces the channel-independence configuration (recently introduced in PatchTST [24]), which processes each channel individually and provides greater flexibility in handling cross-domain time series.

## 4 THE UNITIME MODEL

In this section, we present the proposed UniTime model, an innovative and generic solution designed for end-to-end learning with cross-domain time series data. Figure 2 provides an overview of the UniTime model, which comprises three primary components: a time series tokenizer to preprocess time series raw signals and prepare the time series tokens, a Language-TS Transformer for domain identification and the alignment of two modalities (text and time series), and a decoder for prediction generation. Given our adoption of the channel-independence setting, we next offer a detailed description of each model component from the perspective of a univariate time series from an arbitrary application domain. Formally, we denote the $i$-th univariate time series from domain $\tau$ with length $L_\tau$ as $\boldsymbol{x}_{\tau,i}^{L_\tau} = \{x_{\tau,i}^1, ..., x_{\tau,i}^{L_\tau}\} \in \mathbb{R}^{L_\tau}$.

### 4.1 Time Series Tokenizer

We propose a time series tokenizer to generate the time series tokens from raw series signals. These tokens will be fed into the proposed Language-TS Transformer, described in next section. Our time series tokenizer involves two sub-modules.

**Time Series Patching.** Recognizing that individual time points lack sufficient semantic meaning like a word in a sentence, we employ patching techniques, as seen in ViT [5] and PatchTST [24], to aggregate adjacent time series into tokens. This helps capture local semantic information in time series, and also reduces the computational overhead when processing long input sequences.

Before patching, we preprocess the raw time series through three steps: (1) masking by a binary vector containing zeros and ones (explained later), (2) series stationarization to mitigate distribution shifts [20, 33], and (3) series padding, which involves duplicating the last value of the original sequence to ensure proper patching. We then segment each univariate time series $\boldsymbol{x}_{\tau,i}^{L_\tau}$ into tokens, which may or may not overlap each other, depending on the specific choice. Concretely, let $P$ denote the time series token length and $S_\tau$ represent the stride value (the non-overlapping distance between the starting point of two consecutive tokens). The patching process generates a sequence of tokens $X_{\tau,i}^{N_\tau} \in \mathbb{R}^{N_\tau \times P}$, where $N_\tau$ is the resulting number of tokens, and $N_\tau = \lceil \frac{L_\tau - P}{S_\tau} \rceil + 1$.

We then employ a shared and learnable linear projection to embed the tokens of each domain to a hidden space $Z_{\tau,i}^{N_\tau} \in \mathbb{R}^{N_\tau \times D}$, where $D$ is set to match that of the Transformer used later. It is worth mentioning that the token size $P$ is fixed and shared across domains due to the usage of the linear projection. The stride value $S_\tau$, on the other hand, is adaptable and depends on the historical observation lengths in each domain.

**Masking & Gated Fusion.** Different time series domains manifest varying convergence rates due to their inherent characteristics. For example, domains with simple and regular patterns may converge swiftly, followed by a tendency to overfit, while others may demand more iterations to achieve convergence. Such an imbalanced learning process results in compromised cross-domain forecasting performance. To alleviate this problem, we propose to employ masking to compel the model to depend only on partial input. Consequently, the model is constrained from learning trivial solutions (e.g., simply memorizing the exclusive patterns of data) on domains

that are prone to overfitting, promoting the acquisition of more robust and generalizable representations.

Concretely, for each time series channel, we first generate a binary mask vector $m_{\tau,i}^{L_\tau} \in \{0,1\}^{L_\tau}$, where the value 0 indicates the specific time steps to be masked, and the ratio of zeros is specified by a parameter $r_m$. This mask vector has two usages: (1) masking the raw time series signals $x_{\tau,i}^{L_\tau}$, and (2) serving as a binary indicator to make the model aware of which positions are masked. To achieve the second usage, the mask vector needs to undergo a process similar to that of the time series signals, i.e., padding and patching. Subsequently, we apply a linear projection to map it into the hidden space, denoted by $M_{\tau,i}^{N_\tau} \in \mathbb{R}^{N_\tau \times D}$. Then we perform a gated fusion operation to integrate its information with the time series tokens, in order to enhance the model's awareness of which specific information can be used to generate the predictions. Formally,

$$Z_{\tau,i}^{N_\tau} = Gate \odot Z_{\tau,i}^{N_\tau} + (1 - Gate) \odot M_{\tau,i}^{N_\tau} \tag{1}$$

$$Gate = \sigma(Z_{\tau,i}^{N_\tau} W_{g1} + M_{\tau,i}^{N_\tau} W_{g2} + b_g) \tag{2}$$

where $W_{g1}, W_{g2}, b_g$ are learnable parameters and $\sigma(\cdot)$ is a sigmoid function.

## 4.2 Language-TS Transformer

**Motivation.** When training a model across time series domains, especially when these domains exhibit significant differences in temporal patterns or distributions [20, 34], the model may encounter challenges in distinguishing and generalizing between them. This issue, which we refer to as *domain confusion*, leads to poor forecasting performance in our empirical evaluations.

In this study, we propose the use of domain instructions to offer explicit domain identification information to the model, facilitating the model to discern the source of each time series and adapt its forecasting strategy accordingly. The domain instructions are essentially sentences describing each domain's data. They are also crafted by humans to incorporate human prior knowledge of the data. For example, instructions can be written in a similar form to help the model recognize that the time series of these domains tend to be similar. Moreover, we propose the use of a Language-TS Transformer to learn joint representations from domain instructions and time series, which enables cross-domain generalization by aligning the time series from various input spaces to the common latent space of the language models.

**Model Design.** In this study, we leverage a pretrained language model to unify language and time series modalities. It is important to note that various language models with different architectures are available, including BERT [14], T5 [26], and GPT2 [25]. Given the autoregressive nature of time series data, we opt for GPT2 as our backbone model, which employs causal masking to preserve the temporal order of inputs. Moreover, it is crucial to consider the order of language and time series when using causal masking. If we place the time series data first, the Transformer won't have access to the domain instructions while processing the time series. This weakens the utility of the text information. Therefore, we choose to position the instructions before the time series data, enabling the model to directly leverage contextual identifiers to enhance its cross-domain forecasting performance.

Formally, let $e_\tau$ denote the instruction from domain $\tau$ with length $I_\tau$ and $E_{\tau,i}^{I_\tau} \in \mathbb{R}^{I_\tau \times D}$ denote its embeddings. The input to the proposed Language-TS Transformer is: $H_{\tau,i}^{I_\tau+N_\tau} = \left(E_{\tau,i}^{I_\tau}||Z_{\tau,i}^{N_\tau}\right) + W_{pos}$, where $||$ represents the concatenation operation, and $W_{pos}$ is the learnable positional embeddings from the pretrained language model. Kindly note that the first dimension of $H_{\tau,i}^{I_\tau+N_\tau} \in \mathbb{R}^{(I_\tau+N_\tau) \times D}$ varies across domains. This variability is feasible due to Transformer's capability to handle inputs of different lengths. Then we feed $H_{\tau,i}^{I_\tau+N_\tau}$ into $L_{lm}$ Transformer layers with causal attention, whose weights are initialized from GPT2 [25]. We change the superscript of $H_{\tau,i}^{I_\tau+N_\tau}$ to denote the layer index temporarily, and for layer $l = 1, ..., L_{lm}$, the forward process is:

$$\tilde{H}_{\tau,i}^{l-1} = \text{LN}(\text{MSA}(H_{\tau,i}^{l-1})) + H_{\tau,i}^{l-1} \tag{3}$$

$$H_{\tau,i}^{l} = \text{LN}(\text{MLP}(\tilde{H}_{\tau,i}^{l-1})) + \tilde{H}_{\tau,i}^{l-1} \tag{4}$$

where LN, MSA, and MLP denote a layer normalization, a multi-head self-attention, and a multi-layer perceptron, respectively. Within the MSA, the causal attention is formalized as:

$$\text{Attention}(H_{\tau,i}^{l-1}) = \text{softmax}(\frac{Q^l(K^l)^T}{\sqrt{d_k}} + \text{C})V^l \tag{5}$$

$$\text{C} = \begin{cases} 0, & \text{if position } i \text{ is before } j \\ -\infty, & \text{otherwise} \end{cases} \tag{6}$$

where $Q^l, K^l, V^l$ are the query, key, and value matrices at layer $l$ derived from $H_{\tau,i}^{l-1}$, $d_k$ is the dimension of key, and C is a causal mask matrix.

## 4.3 Decoder

Employing a linear layer to directly produce long-term forecasting results has demonstrated great promise [24, 33, 37], outperforming the traditional iterative approach that is susceptible to substantial error accumulation effects. However, recall that the output of the Language-TS Transformer $H_{\tau,i}^{I_\tau+N_\tau} \in \mathbb{R}^{(I_\tau+N_\tau) \times D}$, which serves as the input to the linear layer, exhibits variations in token lengths. Moreover, the predictive lengths can also vary significantly across diverse domains. These two sources of variability pose a challenge, making it impractical to apply the linear layer directly.

To address this problem, we introduce a maximum token length parameter $R$ and initialize a learnable padding token to ensure consistent sequence lengths across domains. Specifically, we append the padding token repeatedly to $H_{\tau,i}^{I_\tau+N_\tau}$ until the sequence reaches the length of $R$. Then we employ a lightweight Transformer with $L_{light}$ ($L_{light} \ll L_{lm}$) layers to process the padding result. This step serves to inform the other tokens about the presence of the padding token. Finally, we flatten the lightweight Transformer output $\bar{H}_{\tau,i}^{R} \in \mathbb{R}^{R \times D}$ and utilize a linear layer with a maximum predictive length parameter $O$ to generate predictions. The entire procedure is formalized as follows:

$$\bar{H}_{\tau,i}^{R} = \text{LightTrans}(\text{Pad}(H_{\tau,i}^{I_\tau+N_\tau})) \tag{7}$$

$$\hat{x}_{\tau,i}^{O} = \text{Linear}(\text{Flatten}(\bar{H}_{\tau,i}^{R})) \tag{8}$$

Note that our model will always generate $O$ values during forecasting. For domains whose predictive length $T_\tau$ is less than $O$, we truncate the first $T_\tau$ values in $\hat{x}^O_{\tau,i}$ as the forecasting outcomes.

## 4.4 Model Training

**Training Objective.** We utilize the widely used mean squared error to assess the disparity between the prediction and the ground truth. Moreover, we simultaneously predict future values and reconstruct past histories, encouraging the model to align its predictions with the observed historical trends [2]. The overall objective loss in domain $\tau$ is averaged over $c_\tau$ channels, and we get:

$$\mathcal{L}_\tau = \frac{1}{c_\tau} \sum_{i=1}^{c_\tau} \left( \frac{1}{T_\tau} ||\hat{x}^{T_\tau}_{\tau,i} - x^{T_\tau}_{\tau,i}||^2_2 + \frac{1}{L_\tau} ||\hat{x}^{L_\tau}_{\tau,i} - x^{L_\tau}_{\tau,i}||^2_2 \right) \quad (9)$$

**Training Process.** A straightforward approach to cross-domain training involves sequentially feeding each domain's training set to the model during each epoch. However, this method often results in unstable learning and the issue of catastrophic forgetting [6]. To mitigate this problem, we adopt a more granular approach – operating at the batch level. To be specific, we construct batches of data by randomly selecting instances from a pool that encompasses all training data of all involved time series domains. But note that each batch only consists of the data from a single domain. This restriction is due to the varying channel numbers and sequence lengths of each domain. Furthermore, we employ oversampling techniques for domains that have significantly fewer training samples than others. By doing so, we ensure that the model receives ample exposure to these underrepresented domains, preventing them from being overshadowed by the more abundant ones. More details are provided in Appendix A.

## 5 EXPERIMENTS

### 5.1 Experimental Setup

**Datasets.** We extensively assess the proposed UniTime model on eight real-world benchmark datasets, which cover various time series application domains. Here are brief descriptions of the data: (1) **ETT** [40] contains factors used for monitoring electricity transformers between July 2016 and July 2018. These factors include six power load features and readings of oil temperature. ETT involves four subsets. ETTm1 and ETTm2 are recorded at 15-minute intervals, while ETTh1 and ETTh2 are recorded hourly. (2) **Electricity**[1] comprises hourly power consumption of 321 clients from 2012 to 2014. (3) **Exchange** [17] records daily exchange rates of eight different countries ranging from 1990 to 2016. (4) **Weather**[2] is recorded every 10 minutes in the year of 2020. It contains 21 meteorological indicators, such as temperature, humidity, and precipitation. (5) **Illness**[3] includes weekly recorded data on the number of patients with seven influenza-like illnesses between 2002 and 2021. Table 1 provides a summary of the datasets. It can be seen that time series data from various domains exhibit differences in terms of the number of variables, the semantics of those variables, the sampling frequency, and the size of the collected data.

[1]https://archive.ics.uci.edu/ml/datasets/ ElectricityLoadDiagrams20112014.

[2]https://www.bgc-jena.mpg.de/wetter/

[3]https://gis.cdc.gov/grasp/fluview/fluportaldashboard.html

**Table 1: Summary of datasets.**

| Dataset Name | #Variable | Frequency | #Instances | Application Domain |
|---|---|---|---|---|
| ETTm1/ETTm2 | 7 | 15 mins | 57,507 | Electrical Asset Monitoring |
| ETTh1/ETTh2 | 7 | 1 hour | 14,307 | Electrical Asset Monitoring |
| Electricity | 321 | 1 hour | 26,211 | Electricity Consumption |
| Weather | 21 | 10 mins | 52,603 | Meteorologic Monitoring |
| Exchange | 8 | 1 day | 7,207 | Foreign Exchange Market |
| Illness | 7 | 1 week | 861 | Epidemiological Monitoring |

**Baselines.** We include eight state-of-the-art methods for multivariate time series forecasting comparisons, including Informer [40], Autoformer [34], FEDformer [41], NSformer [20], DLinear [37], TimesNet [33], PatchTST [24], and GPT4TS [42], a recent paper that uses language models to process time series data. Note that all these methods train a dedicated model for each evaluated dataset and for each assessed predictive length in their original papers.

**Implementation Details.** We integrate our method into the established pipeline[4] from Wu et al. [33], which serves as a robust evaluation platform for various baseline methods. We also adhere to the same experimental settings as in Wu et al. [33] to ensure a fair comparison: we set the maximum number of epochs to 10 and fix the lookback window length to 36 for the Illness dataset, and 96 for the others. Moreover, we utilize a pretrained GPT2 [25] model as the backbone, with its layer count $L_{lm}$ set at 6, and we do not freeze any of its parameters. For the lightweight Transformer, we configure the $L_{light}$ to 2. The patch length $P$, maximum token length $R$, maximum predictive length $O$, mask ratio $r_m$ are consistently set to 16, 17, 720, and 0.5, respectively. The configuration specifics for each dataset and the results of the hyperparameter studies are provided in Appendix A and B. We train our method via the AdamW optimizer with an initial learning rate of 0.0001. Regarding model selection, we calculate the validation loss for all the datasets involved and then compute an average score. The model that achieves the lowest overall validation loss will be used for testing. Experiments are repeated three times with different seeds on an NVIDIA A100 GPU. We implement UniTime using PyTorch 1.12, and *will release the code on GitHub for public use.*

### 5.2 Main Results

Table 2 presents the overall forecasting performance. We utilize two vertical lines to demarcate the table. The right part of the table signifies that separate models are trained for each dataset and for each specific predictive length. To illustrate, for the ETTm1 dataset, four distinct models are created to predict four different future lengths: 96, 192, 336, and 720. On the left side of the table, models are trained across datasets and consistently generate 720 future values. When evaluating performance for a setting shorter than 720 entries, such as 96, we simply take the first 96 values within the 720-value output. According to the table, the proposed UniTime model demonstrates remarkable improvements over the baseline models that are also trained across datasets, securing the best performance in 79 out of 80 entries. Moreover, UniTime delivers competitive results when compared to models trained individually on each dataset, as demonstrated by improving 37 out of 80 entries to the new state-of-the-art. This outcome validates the effectiveness of our model in handling time series data with diverse characteristics, such as sampling frequency and periodicity.

[4]https://github.com/thuml/Time-Series-Library

**Table 2: Forecasting performance comparisons. The input sequence length is set to 36 for the Illness dataset and 96 for the others. The predictive lengths are set to {24, 36, 48, 60} for Illness, and {96, 192, 336, 720} for others. Avg is averaged over all predictive lengths. Note that we bold the best performance among models trained across datasets, which is on the left-hand side of the two vertical lines, and we bold and underline the best performance for the entire row.**

| | Method | Models Trained Across Datasets | | | | | | Models Trained on Each Dataset | | | | | | | | | | | | | | | |
|---|---|---|---|---|---|---|---|---|---|---|---|---|---|---|---|---|---|---|---|---|---|---|---|
| | | UniTime | | GPT4TS† | | PatchTST† | | GPT4TS* | | PatchTST* | | TimesNet | | DLinear | | NSformer | | FEDformer | | Autoformer | | Informer | |
| | | MSE | MAE | MSE | MAE | MSE | MAE | MSE | MAE | MSE | MAE | MSE | MAE | MSE | MAE | MSE | MAE | MSE | MAE | MSE | MAE | MSE | MAE |
| ETTm1 | 96 | **0.322** | **0.363** | 0.509 | 0.463 | 0.927 | 0.604 | 0.335 | 0.369 | 0.344 | 0.373 | 0.338 | 0.375 | 0.345 | 0.372 | 0.386 | 0.398 | 0.379 | 0.419 | 0.505 | 0.475 | 0.672 | 0.571 |
| | 192 | **0.366** | **0.387** | 0.537 | 0.476 | 0.964 | 0.620 | 0.374 | **0.385** | 0.367 | 0.386 | 0.374 | 0.387 | 0.380 | 0.389 | 0.459 | 0.444 | 0.426 | 0.441 | 0.553 | 0.496 | 0.795 | 0.669 |
| | 336 | **0.398** | **0.407** | 0.564 | 0.488 | 1.041 | 0.656 | 0.407 | **0.406** | **0.392** | 0.407 | 0.410 | 0.411 | 0.413 | 0.413 | 0.495 | 0.464 | 0.445 | 0.459 | 0.621 | 0.537 | 1.212 | 0.871 |
| | 720 | **0.454** | **0.440** | 0.592 | 0.504 | 0.950 | 0.636 | 0.469 | 0.442 | 0.464 | 0.442 | 0.478 | 0.450 | 0.474 | 0.453 | 0.585 | 0.516 | 0.543 | 0.490 | 0.671 | 0.561 | 1.166 | 0.823 |
| | Avg | **0.385** | **0.399** | 0.551 | 0.483 | 0.971 | 0.629 | 0.396 | 0.401 | 0.392 | 0.402 | 0.400 | 0.406 | 0.403 | 0.407 | 0.481 | 0.456 | 0.448 | 0.452 | 0.588 | 0.517 | 0.961 | 0.734 |
| ETTm2 | 96 | **0.183** | **0.266** | 0.229 | 0.304 | 0.240 | 0.318 | 0.190 | 0.275 | **0.177** | **0.260** | 0.187 | 0.267 | 0.193 | 0.292 | 0.192 | 0.274 | 0.203 | 0.287 | 0.255 | 0.339 | 0.365 | 0.453 |
| | 192 | **0.251** | **0.310** | 0.287 | 0.338 | 0.301 | 0.352 | 0.253 | 0.313 | **0.246** | **0.305** | 0.249 | 0.309 | 0.284 | 0.362 | 0.280 | 0.339 | 0.269 | 0.328 | 0.281 | 0.340 | 0.533 | 0.563 |
| | 336 | **0.319** | **0.351** | 0.337 | 0.367 | 0.367 | 0.391 | 0.321 | 0.360 | **0.305** | **0.343** | 0.321 | 0.351 | 0.369 | 0.427 | 0.334 | 0.361 | 0.325 | 0.366 | 0.339 | 0.372 | 1.363 | 0.887 |
| | 720 | **0.420** | **0.410** | 0.430 | 0.416 | 0.451 | 0.432 | 0.411 | 0.406 | 0.410 | 0.405 | **0.408** | **0.403** | 0.554 | 0.522 | 0.417 | 0.413 | 0.421 | 0.415 | 0.433 | 0.432 | 3.379 | 1.338 |
| | Avg | **0.293** | **0.334** | 0.321 | 0.356 | 0.340 | 0.373 | 0.294 | 0.339 | **0.285** | **0.328** | 0.291 | 0.333 | 0.350 | 0.401 | 0.306 | 0.347 | 0.305 | 0.349 | 0.327 | 0.371 | 1.410 | 0.810 |
| ETTh1 | 96 | **0.397** | 0.418 | 0.449 | 0.424 | 0.409 | **0.403** | 0.398 | 0.424 | 0.404 | 0.413 | 0.384 | 0.402 | 0.386 | **0.400** | 0.513 | 0.491 | **0.376** | 0.419 | 0.449 | 0.459 | 0.865 | 0.713 |
| | 192 | **0.434** | **0.439** | 0.503 | 0.453 | 0.467 | 0.444 | 0.449 | **0.427** | 0.454 | 0.440 | 0.436 | 0.429 | 0.437 | 0.432 | 0.534 | 0.504 | **0.420** | 0.448 | 0.500 | 0.482 | 1.008 | 0.792 |
| | 336 | **0.468** | **0.457** | 0.540 | 0.477 | 0.509 | 0.472 | 0.492 | 0.466 | 0.497 | 0.462 | 0.491 | 0.469 | 0.481 | 0.459 | 0.588 | 0.535 | **0.459** | 0.465 | 0.521 | 0.496 | 1.107 | 0.809 |
| | 720 | **0.469** | **0.477** | 0.515 | 0.489 | 0.503 | 0.485 | 0.487 | 0.483 | 0.496 | 0.481 | 0.521 | 0.500 | 0.519 | 0.516 | 0.643 | 0.616 | 0.506 | 0.507 | 0.514 | 0.512 | 1.181 | 0.865 |
| | Avg | **0.442** | **0.448** | 0.502 | 0.461 | 0.472 | 0.451 | 0.457 | 0.450 | 0.463 | 0.449 | 0.458 | 0.450 | 0.456 | 0.452 | 0.570 | 0.537 | **0.440** | 0.460 | 0.496 | 0.487 | 1.040 | 0.795 |
| ETTh2 | 96 | **0.296** | **0.345** | 0.303 | 0.349 | 0.314 | 0.361 | 0.312 | 0.360 | 0.312 | 0.358 | 0.340 | 0.374 | 0.333 | 0.387 | 0.476 | 0.458 | 0.358 | 0.397 | 0.346 | 0.388 | 3.755 | 1.525 |
| | 192 | **0.374** | **0.394** | 0.391 | 0.399 | 0.407 | 0.411 | 0.387 | 0.405 | 0.397 | 0.408 | 0.402 | 0.414 | 0.477 | 0.476 | 0.512 | 0.493 | 0.429 | 0.439 | 0.456 | 0.452 | 5.602 | 1.931 |
| | 336 | **0.415** | **0.427** | 0.422 | 0.428 | 0.437 | 0.443 | 0.424 | 0.437 | 0.435 | 0.440 | 0.452 | 0.452 | 0.594 | 0.541 | 0.552 | 0.551 | 0.496 | 0.487 | 0.482 | 0.486 | 4.721 | 1.835 |
| | 720 | **0.425** | **0.444** | 0.429 | 0.449 | 0.434 | 0.448 | 0.433 | 0.453 | 0.436 | 0.449 | 0.462 | 0.468 | 0.831 | 0.657 | 0.562 | 0.560 | 0.463 | 0.474 | 0.515 | 0.511 | 3.647 | 1.625 |
| | Avg | **0.378** | **0.403** | 0.386 | 0.406 | 0.398 | 0.416 | 0.389 | 0.414 | 0.395 | 0.414 | 0.414 | 0.427 | 0.559 | 0.515 | 0.526 | 0.516 | 0.437 | 0.449 | 0.450 | 0.459 | 4.431 | 1.729 |
| Electricity | 96 | **0.196** | **0.287** | 0.232 | 0.321 | 0.198 | 0.290 | 0.197 | 0.290 | 0.186 | **0.269** | **0.168** | 0.272 | 0.197 | 0.282 | 0.169 | 0.273 | 0.193 | 0.308 | 0.201 | 0.317 | 0.274 | 0.368 |
| | 192 | **0.199** | **0.291** | 0.234 | 0.325 | 0.202 | 0.293 | 0.201 | 0.292 | 0.190 | **0.273** | 0.184 | 0.289 | 0.196 | 0.285 | **0.182** | 0.286 | 0.201 | 0.315 | 0.222 | 0.334 | 0.296 | 0.386 |
| | 336 | **0.214** | **0.305** | 0.249 | 0.338 | 0.223 | 0.318 | 0.217 | 0.309 | 0.206 | **0.290** | **0.198** | 0.300 | 0.209 | 0.301 | 0.200 | 0.304 | 0.214 | 0.329 | 0.231 | 0.338 | 0.300 | 0.394 |
| | 720 | **0.254** | **0.335** | 0.289 | 0.366 | 0.259 | 0.341 | 0.253 | 0.339 | 0.247 | 0.322 | **0.220** | **0.320** | 0.245 | 0.333 | 0.222 | 0.321 | 0.246 | 0.355 | 0.254 | 0.361 | 0.373 | 0.439 |
| | Avg | **0.216** | **0.305** | 0.251 | 0.338 | 0.221 | 0.311 | 0.217 | 0.308 | 0.207 | **0.289** | **0.192** | 0.295 | 0.212 | 0.300 | 0.193 | 0.296 | 0.214 | 0.327 | 0.227 | 0.338 | 0.311 | 0.397 |
| Weather | 96 | **0.171** | **0.214** | 0.212 | 0.251 | 0.213 | 0.260 | 0.203 | 0.244 | 0.177 | 0.218 | 0.172 | 0.220 | 0.196 | 0.255 | 0.173 | 0.223 | 0.217 | 0.296 | 0.266 | 0.336 | 0.300 | 0.384 |
| | 192 | **0.217** | **0.254** | 0.261 | 0.288 | 0.269 | 0.300 | 0.247 | 0.277 | 0.222 | 0.259 | 0.219 | 0.261 | 0.237 | 0.296 | 0.245 | 0.285 | 0.276 | 0.336 | 0.307 | 0.367 | 0.598 | 0.544 |
| | 336 | **0.274** | **0.293** | 0.313 | 0.324 | 0.330 | 0.341 | 0.297 | 0.311 | 0.277 | 0.297 | 0.280 | 0.306 | 0.283 | 0.335 | 0.321 | 0.338 | 0.339 | 0.380 | 0.359 | 0.395 | 0.578 | 0.523 |
| | 720 | **0.351** | **0.343** | 0.386 | 0.372 | 0.404 | 0.389 | 0.368 | 0.356 | 0.352 | 0.347 | 0.365 | 0.359 | **0.345** | 0.381 | 0.414 | 0.410 | 0.403 | 0.428 | 0.419 | 0.428 | 1.059 | 0.741 |
| | Avg | **0.253** | **0.276** | 0.293 | 0.309 | 0.304 | 0.323 | 0.279 | 0.297 | 0.257 | 0.280 | 0.259 | 0.287 | 0.265 | 0.317 | 0.288 | 0.314 | 0.309 | 0.360 | 0.338 | 0.382 | 0.634 | 0.548 |
| Exchange | 96 | **0.086** | **0.209** | 0.142 | 0.261 | 0.137 | 0.260 | 0.091 | 0.212 | 0.109 | 0.236 | 0.107 | 0.234 | 0.088 | 0.218 | 0.111 | 0.237 | 0.148 | 0.278 | 0.197 | 0.323 | 0.847 | 0.752 |
| | 192 | **0.174** | **0.299** | 0.224 | 0.339 | 0.222 | 0.341 | 0.183 | 0.304 | 0.205 | 0.327 | 0.226 | 0.344 | 0.176 | 0.315 | 0.219 | 0.335 | 0.271 | 0.380 | 0.300 | 0.369 | 1.204 | 0.895 |
| | 336 | **0.319** | **0.408** | 0.377 | 0.448 | 0.372 | 0.447 | 0.328 | 0.417 | 0.356 | 0.436 | 0.367 | 0.448 | **0.313** | 0.427 | 0.421 | 0.476 | 0.460 | 0.500 | 0.509 | 0.524 | 1.672 | 1.036 |
| | 720 | **0.875** | **0.701** | 0.939 | 0.736 | 0.912 | 0.727 | 0.880 | 0.704 | 0.888 | 0.716 | 0.964 | 0.746 | **0.839** | **0.695** | 1.092 | 0.769 | 1.195 | 0.841 | 1.447 | 0.941 | 2.478 | 1.310 |
| | Avg | **0.364** | **0.404** | 0.421 | 0.446 | 0.411 | 0.444 | 0.371 | 0.409 | 0.390 | 0.429 | 0.416 | 0.443 | **0.354** | 0.414 | 0.461 | 0.454 | 0.519 | 0.500 | 0.613 | 0.539 | 1.550 | 0.998 |
| Illness | 24 | **2.346** | **0.954** | 3.322 | 1.278 | 4.289 | 1.485 | 2.732 | 1.100 | 2.335 | 0.989 | 2.317 | **0.934** | 2.398 | 1.040 | **2.294** | 0.945 | 3.228 | 1.260 | 3.483 | 1.287 | 5.764 | 1.677 |
| | 36 | **1.998** | **0.912** | 3.696 | 1.374 | 4.360 | 1.510 | 2.664 | 1.063 | 2.561 | 1.035 | 1.972 | 0.920 | 2.646 | 1.088 | **1.825** | **0.848** | 2.679 | 1.080 | 3.103 | 1.148 | 4.755 | 1.467 |
| | 48 | **1.979** | **0.912** | 3.765 | 1.402 | 4.209 | 1.481 | 2.617 | 1.041 | 2.465 | 1.022 | 2.238 | 0.940 | 2.614 | 1.086 | 2.010 | **0.900** | 2.622 | 1.078 | 2.669 | 1.085 | 4.763 | 1.469 |
| | 60 | **2.109** | **0.938** | 3.928 | 1.432 | 3.981 | 1.444 | 2.478 | 1.035 | 2.189 | 0.997 | **2.027** | **0.928** | 2.804 | 1.146 | 2.178 | 0.963 | 2.857 | 1.157 | 2.770 | 1.125 | 5.264 | 1.564 |
| | Avg | **2.108** | **0.929** | 3.678 | 1.372 | 4.210 | 1.480 | 2.623 | 1.060 | 2.388 | 1.011 | 2.139 | 0.931 | 2.616 | 1.090 | **2.077** | **0.914** | 2.847 | 1.144 | 3.006 | 1.161 | 5.137 | 1.544 |
| 1st Count | | 37 | | 0 | | 0 | | 3 | | 13 | | 10 | | 6 | | 7 | | 4 | | 0 | | 0 | |

† means that we modify the baselines' code (e.g., use padding to align input lengths across different domains), and make them train and test in the same way as our method. ∗ indicates that we adopt the official code of the baselines and reset their input sequence length and maximum training epochs number for a fair comparison to other methods. Other results are from TimesNet [33].

## 5.3 Ablation Studies

To better understand the effectiveness of model designs in UniTime, we construct five variants of the model and conduct ablation studies across all evaluated datasets. The experimental results are summarized in Table 3, indicating that all the designed components are indispensable.

Firstly, *w/o instructions* causes a significant drop in performance across all datasets, with the most pronounced effects on ETTm1 and Illness. This emphasizes the critical role of domain instructions

in providing identification information to the model. To further investigate the domain confusion issue, we conduct a comparison between the hidden representations of UniTime *w/o instructions* and UniTime *w/ instructions* using the T-SNE visualization tool [27]. Specifically, for each dataset, we randomly select 100 samples from their respective test sets, and visualize the hidden representations produced by the Language-TS Transformer. In Figure 4, we can observe that in the absence of instructions, the representations of different domains are mixed together, whereas with the inclusion of

**Table 3: Ablation of method designs. Due to page limit, for each dataset, we report the average value over all predictive lengths. Full results are provided in Table 7.**

| Variant | ETTm1 | | ETTm2 | | ETTh1 | | ETTh2 | | Electricity | | Weather | | Exchange | | Illness | |
|---|---|---|---|---|---|---|---|---|---|---|---|---|---|---|---|---|
| | MSE | MAE | MSE | MAE | MSE | MAE | MSE | MAE | MSE | MAE | MSE | MAE | MSE | MAE | MSE | MAE |
| UniTime | **0.385** | **0.399** | 0.293 | 0.334 | 0.442 | 0.448 | **0.378** | **0.403** | 0.216 | 0.305 | **0.253** | **0.276** | **0.364** | **0.404** | **2.108** | **0.929** |
| w/o instructions | 0.479 | 0.461 | 0.311 | 0.349 | 0.466 | 0.449 | 0.397 | 0.409 | 0.221 | 0.310 | 0.283 | 0.307 | 0.389 | 0.428 | 2.381 | 1.041 |
| w/o masking | 0.390 | 0.408 | **0.286** | **0.332** | 0.459 | 0.461 | 0.380 | 0.406 | **0.210** | **0.298** | 0.257 | 0.280 | 0.379 | 0.417 | 2.606 | 1.112 |
| w/o LightTrans | 0.392 | 0.402 | 0.295 | 0.336 | 0.443 | **0.445** | 0.382 | 0.405 | 0.222 | 0.308 | 0.261 | 0.284 | 0.375 | 0.414 | 2.303 | 0.998 |
| w/o reconstruction | 0.392 | 0.405 | 0.294 | 0.336 | **0.439** | 0.447 | 0.383 | 0.407 | 0.220 | 0.312 | 0.259 | 0.281 | 0.383 | 0.417 | 2.197 | 0.956 |
| w/o all | 0.487 | 0.462 | 0.313 | 0.352 | 0.469 | 0.459 | 0.391 | 0.407 | 0.219 | 0.308 | 0.276 | 0.297 | 0.395 | 0.430 | 2.479 | 1.084 |

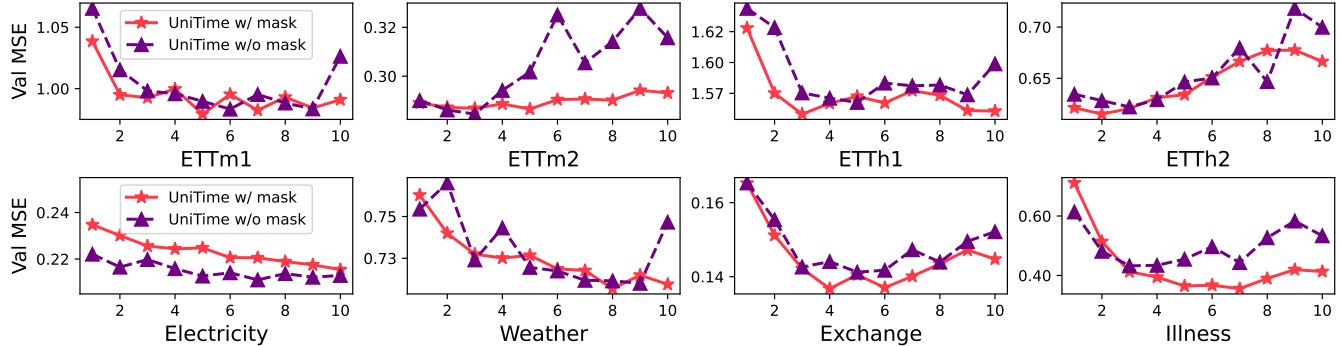

**Figure 3: Visualization of the validation loss during model training. The x-axis denotes the training epoch number.**

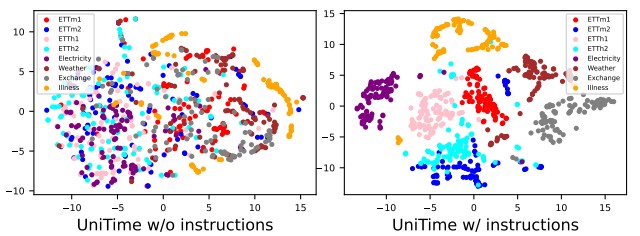

**Figure 4: T-SNE visualization of the hidden representations.**

instructions, they exhibit clear clustering-like patterns. This observation confirms the existence of domain confusion, and underscores the effectiveness of instructions as a tool to address this issue. Note that in the visualization of UniTime *w/ instructions*, the clusters of ETTm1, ETTm2, ETTh1, ETTh2 are close to each other. This proximity is attributed to the fact that they belong to the same domain and thus share underlying temporal characteristics.

The results obtained under the *w/o masking* setting reveal that while the model performs satisfactorily on some datasets, the performance on other datasets is significantly degraded, especially on Illness. This decline can be attributed to an imbalanced cross-domain learning process that occurs when masking is disabled. To illustrate this point further, we have plotted the changes in validation loss in Figure 3. Recall that the overall validation loss across all domains is a critical factor during the model selection process. When masking is turned off, the datasets display varying convergence speeds. For example, ETTm2, ETTh2, Exchange, and Illness experience severe overfitting beyond the 4th epoch, while others require more epochs to reach convergence. This lack of balance poses challenges in the model selection process when aiming to choose a model that performs well across all datasets. However,

when masking is enabled, the majority of loss curves do not demonstrate an overfitting trend. Instead, they converge at a later phase and exhibit increased stability. Such a balanced learning environment allows the model to be selected in the later phase of training, leading to superior overall performance.

Furthermore, *w/o LightTrans* and *w/o reconstruction* mean that we remove the light Transformer after the language model and disable the auxiliary reconstruction loss, respectively. The results show that both of them are effective in boosting the overall performance. Finally, the setting of *w/o all* turns off all the aforementioned designs, resulting in degraded performance across all datasets.

## 5.4 Zero-Shot Transferability Analysis

**Setups.** In this part, we delve into the transferability of our methods and baseline models from the source (training) domains to the target (unseen) domains. Specifically, we first train the models on the datasets of ETTh1, ETTm1, and ETTm2. Then we assess their performance in both in-domain transfer and out-domain transfer scenarios through zero-shot testing. This testing is conducted on ETTh2 (hailing from the same domain as the source), Electricity (a different domain with some underlying relations to the source domain), and Weather (representing a completely unrelated domain).

**Transfer Protocol.** Before executing zero-shot transfers with our UniTime model, a preliminary step involves selecting the appropriate domain instructions for the unseen domain. The rationale behind this is that if two domains share common patterns, they may favor similar instructions for their identification. In this study, we propose an instruction selection protocol that hinges on the instructions visible to the models during training. Specifically, we leverage the model input, namely historical observations, and partition them into two parts: the first part is fed into the model to

**Table 4: Results of design choices related to the language model. We report the average results over all predictive lengths. Full results are offered in Table 7.**

| Variant | ETTm1 | | ETTm2 | | ETTh1 | | ETTh2 | | Electricity | | Weather | | Exchange | | Illness | |
|---|---|---|---|---|---|---|---|---|---|---|---|---|---|---|---|---|
| | MSE | MAE | MSE | MAE | MSE | MAE | MSE | MAE | MSE | MAE | MSE | MAE | MSE | MAE | MSE | MAE |
| UniTime | **0.385** | **0.399** | **0.293** | **0.334** | 0.442 | 0.448 | **0.378** | **0.403** | **0.216** | **0.305** | **0.253** | **0.276** | **0.364** | **0.404** | **2.108** | **0.929** |
| TS-Text | 0.391 | 0.403 | 0.295 | 0.337 | 0.446 | 0.452 | 0.381 | 0.406 | 0.220 | 0.309 | 0.261 | 0.284 | 0.381 | 0.414 | 2.258 | 1.018 |
| Random Init | 0.404 | 0.411 | 0.297 | 0.339 | 0.446 | 0.451 | 0.379 | 0.404 | 0.220 | 0.309 | 0.260 | 0.281 | 0.374 | 0.413 | 2.336 | 1.043 |
| Freeze PLM | 0.398 | 0.410 | 0.297 | 0.338 | 0.444 | 0.452 | 0.378 | 0.405 | 0.224 | 0.314 | 0.262 | 0.283 | 0.373 | 0.409 | 2.481 | 1.078 |
| FPT PLM | 0.391 | 0.407 | 0.295 | 0.336 | **0.438** | **0.446** | 0.378 | 0.403 | 0.220 | 0.310 | 0.260 | 0.283 | 0.376 | 0.412 | 2.286 | 1.028 |

**Table 5: Zero-shot transferability comparisons. Avg is averaged over all predictive lengths.**

| Method | | UniTime | | GPT4TS | | PatchTST | | Repeat | |
|---|---|---|---|---|---|---|---|---|---|
| | | MSE | MAE | MSE | MAE | MSE | MAE | MSE | MAE |
| ETTh2 | 96 | **0.306** | **0.352** | 0.316 | 0.361 | 0.332 | 0.371 | 0.432 | 0.422 |
| | 192 | **0.389** | **0.401** | 0.400 | 0.410 | 0.422 | 0.421 | 0.534 | 0.473 |
| | 336 | **0.424** | **0.434** | 0.430 | 0.439 | 0.462 | 0.455 | 0.597 | 0.511 |
| | 720 | **0.433** | **0.450** | 0.442 | 0.461 | 0.467 | 0.469 | 0.594 | 0.519 |
| | Avg | **0.388** | **0.409** | 0.397 | 0.418 | 0.421 | 0.429 | 0.539 | 0.481 |
| Electricity | 96 | **0.409** | **0.481** | 0.448 | 0.520 | 0.529 | 0.562 | 1.588 | 0.945 |
| | 192 | **0.410** | **0.484** | 0.443 | 0.517 | 0.507 | 0.550 | 1.596 | 0.951 |
| | 336 | **0.439** | **0.504** | 0.462 | 0.526 | 0.536 | 0.566 | 1.618 | 0.961 |
| | 720 | **0.487** | **0.531** | 0.494 | 0.542 | 0.563 | 0.581 | 1.647 | 0.975 |
| | Avg | **0.436** | **0.500** | 0.462 | 0.526 | 0.534 | 0.565 | 1.612 | 0.958 |
| Weather | 96 | **0.210** | 0.262 | 0.223 | 0.271 | 0.235 | 0.277 | 0.259 | **0.254** |
| | 192 | **0.264** | 0.303 | 0.287 | 0.319 | 0.293 | 0.320 | 0.309 | **0.292** |
| | 336 | **0.326** | **0.334** | 0.347 | 0.357 | 0.351 | 0.356 | 0.376 | 0.338 |
| | 720 | **0.402** | **0.382** | 0.432 | 0.409 | 0.427 | 0.404 | 0.465 | 0.394 |
| | Avg | **0.301** | **0.320** | 0.322 | 0.339 | 0.327 | 0.339 | 0.352 | 0.320 |

generate the predictions and the second part is utilized to compute the forecasting loss. This loss calculation offers insights into which instruction is most suitable for the unseen data. Experimentally, we conduct this protocol on 0.5% of test samples to determine the instructions to be used. We then apply the selected instruction to the remaining samples in the test set.

**Results.** Table 5 displays the results of zero-shot testing, with the last column labeled "Repeat" serving as a baseline that simply utilizes the last value of histories as the forecast value for all future time steps. The table clearly illustrates that UniTime consistently outperforms the baselines across the majority of cases, affirming the effectiveness of incorporating instructions. Furthermore, in accordance with our instruction selection protocol, all three zero-shot datasets opt for instructions derived from the data of ETTh1. This choice is well-founded, particularly for the ETTh2 dataset, as it exhibits strong connections with ETTh1. The reason for the Electricity and Weather datasets opting for ETTh1's instruction likely stems from their similar underlying patterns, which lends further support to our approach's adaptability across diverse domains.

## 5.5 Exploration Studies on Language Models

In this section, we conduct further investigations into the factors associated with the language model, aiming to enhance the comprehension of the design choices made in UniTime.

**Input Order.** We explore the effects of altering the input order by placing the time series data before the instructions. In this configuration, time series tokens are unable to attend to the instruction tokens due to the presence of a causal mask. As shown in the second row of Table 4, it's clear that UniTime outperforms this variant with the changed order. The relatively small performance gap is due to our use of a decoder following the Language-TS Transformer. This decoder uses information from the instruction tokens to generate predictions, mitigating the impact of the altered input order.

**Initialization.** In this setting, we forego the use of pretrained weights from GPT-2, opting instead for randomly initialized weights. As evident from the third row of Table 4, we can see that the performance of this configuration on all datasets is inferior to that of our default model. This observation indicates the superiority of pretrained weights, which have been learned from a vast language corpus, in effectively processing textual information.

**Tunability.** In our main results, we fully tuned the pretrained language model (PLM). In this part, we explore alternative approaches: freezing the entire language model, referred to as "Freeze PLM", and freezing the majority of parameters in the language model, denoted as "FPT PLM" [22, 42]. To be specific, the FPT method tunes only the positional embeddings and layer normalization components of the model while keeping the other components, such as self-attention and feed-forward networks, frozen.

The experimental results are summarized in the last two rows of Table 4. Firstly, it is evident that fully tuning the model yields the best performance, followed by the cases of FPT and Freeze. Secondly, a noteworthy finding is that the performance remains relatively strong even when we freeze the entire language model. This outcome suggests that the language model possesses the capability to process time series tokens and generate reasonable hidden representations. This interesting phenomenon is also observed by a recent study [42], and they attribute such universal computing ability to the self-attention modules of a trained Transformer, which behaves similarly to principal component analysis. Thirdly, considering that only a minor subset of parameters requires tuning under the FPT method, it strikes a good balance between performance and efficiency. This makes it an attractive choice when computational resources are limited.

## 6 CONCLUSION

This paper delves into an innovative and pivotal learning paradigm: developing a unified forecasting model capable of accommodating diverse time series application domains. We identify the challenges in constructing such a unified model and propose the novel UniTime to address them accordingly. Our extensive evaluations confirm the effectiveness of UniTime compared to existing solutions. We believe that this work represents a significant step towards building a foundation model for general time series forecasting.

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

# A MORE CONFIGURATION DETAILS

This section offers detailed configurations for each dataset evaluated in this study. The results are shown in Table 6. First, we partition all datasets into training, validation and test set in chronological order. The split ratio is 6:2:2 for the ETT series dataset and 7:1:2 for others. We can observe that the datasets ETTm1, ETTm2, and Weather have the highest number of training samples, each exceeding 30,000. They are followed by ETTh1 and ETTh2 with approximately 8,500 samples, Exchange with 5,000 samples, and Illness, which has only 600 samples. Then we determine the batch size for each dataset

**Table 6: Details of the training, validation, and testing set partitions, as well as the configurations specific to different domains.**

| Dataset | #Training | #Validation | #Testing | Batch Size | Oversample Times | Stride | Domain Instructions |
|---|---|---|---|---|---|---|---|
| ETTm1 | 34,465 | 11,521 | 11,521 | 64 | 0 | 16 | Electricity transformer A data with fifteen minutes sample rate. |
| ETTm2 | 34,465 | 11,521 | 11,521 | 64 | 0 | 16 | Electricity transformer B data with fifteen minutes sample rate. |
| ETTh1 | 8,545 | 2,881 | 2,881 | 32 | 0 | 16 | Electricity transformer A data with one hour sample rate. |
| ETTh2 | 8,545 | 2,881 | 2,881 | 32 | 0 | 16 | Electricity transformer B data with one hour sample rate. |
| Electricity | 18,317 | 2,633 | 5,261 | 24 | 0 | 16 | Power consumption data with one hour sample rate. |
| Weather | 36,792 | 5,271 | 10,540 | 64 | 0 | 16 | Meteorological indicator data with ten minutes sample rate. |
| Exchange | 5,120 | 665 | 1,422 | 24 | 0 | 16 | Exchange rate data with one day sample rate. |
| Illness | 617 | 74 | 170 | 16 | 12 | 4 | Patient number data with one week sample rate. |

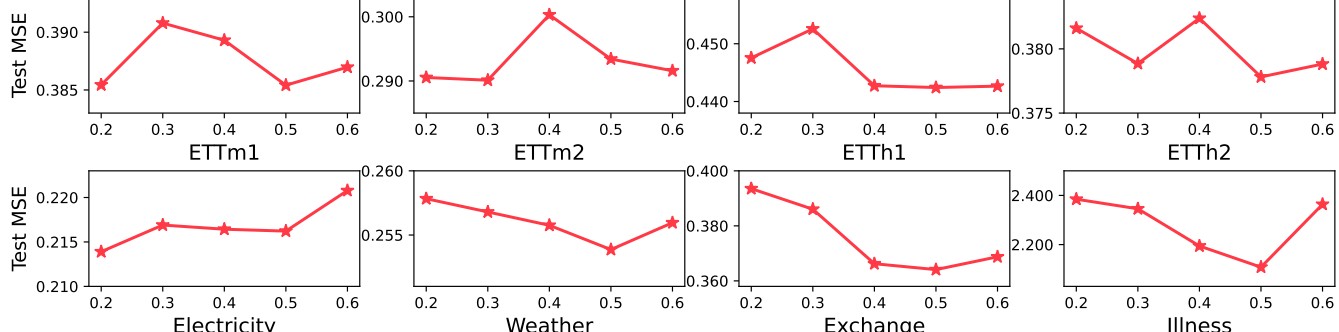

**Figure 5: Effects of mask ratio. The y-axis is the average test MSE over four predictive lengths.**

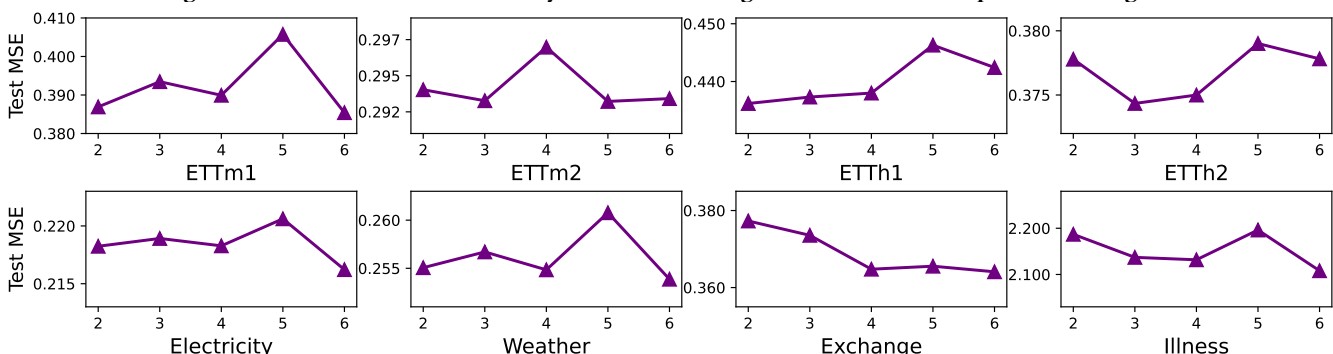

**Figure 6: Effects of Language-TS Transformer's number of layers. The y-axis is the average test MSE over four predictive lengths.**

based on the number of training samples. The guiding principle is to allocate a larger batch size for datasets with more training samples and a smaller batch size for those with fewer samples. This strategy allows the model to undergo more frequent updates when training on smaller datasets during each epoch. Following this principle, we assign a batch size of 64 to the ETTm1, ETTm2, and Weather datasets, 32 to ETTh1 and ETTh2, 24 to the Exchange dataset, and 16 to the Illness dataset. An exception to this principle is for the Electricity dataset, which is supposed to be set to 32, but due to GPU memory constraints, it is set to 24 in our experiments. Furthermore, recall that we implement oversampling to augment the size of datasets with significantly fewer training samples. This strategy is applied to the Illness dataset, which contains only 600 samples. The decision to perform oversampling 12 times on the Illness dataset is based on our empirical assessments. In short, the primary goal of the two strategies is to ensure that the model obtains ample exposure to the underrepresented domains, preventing them from being marginalized by the more abundant ones.

We next elaborate on the design rationales of the domain instructions, which are listed in Table 6. These instructions are essentially sentences that describe the data in each domain. In this study, we aim to keep them concise to reduce the computational and memory cost, considering that we are tuning language models, which typically have a substantial number of parameters. We also try to incorporate human prior knowledge of the data. For instance, we structure the instructions to resemble each other in cases where domains share similarities. Consider the situation of ETTm1, ETTm2, ETTh1, ETTh2: these four datasets originate from the same source. Moreover, ETTm1 and ETTh1, as well as ETTm2 and ETTh2, are essentially the same dataset with different sampling frequencies. Therefore, we craft the instructions for these datasets in a similar form, with differences limited to the identifier of the data, i.e., "A" and "B," and the sampling rate, i.e., "fifteen minutes" and "one hour".

## B  HYPERPARAMETER STUDIES

In this part, we conduct an investigate into two critical hyperparameters: the mask ratio $r_m$ and the number of layers $L_{lm}$ in the Language-TS Transformer. The results of these assessments are depicted in Figure 5 and Figure 6. Regarding the mask ratio value, we observe that the model generally performs better with a larger

**Table 7: Full results of ablation and exploration studies. The predictive lengths are set to $L_1, L_2, L_3, L_4 = \{24, 36, 48, 60\}$ for the Illness dataset, and $L_1, L_2, L_3, L_4 = \{96, 192, 336, 720\}$ for the others.**

| Method | | ETTm1 | | ETTm2 | | ETTh1 | | ETTh2 | | Electricity | | Weather | | Exchange | | Illness | |
|---|---|---|---|---|---|---|---|---|---|---|---|---|---|---|---|---|---|
| | | MSE | MAE | MSE | MAE | MSE | MAE | MSE | MAE | MSE | MAE | MSE | MAE | MSE | MAE | MSE | MAE |
| UniTime | $L_1$ | 0.322 | 0.363 | 0.183 | 0.266 | 0.397 | 0.418 | 0.296 | 0.345 | 0.196 | 0.287 | 0.171 | 0.214 | 0.086 | 0.209 | 2.346 | 0.954 |
| | $L_2$ | 0.366 | 0.387 | 0.251 | 0.310 | 0.434 | 0.439 | 0.374 | 0.394 | 0.199 | 0.291 | 0.217 | 0.254 | 0.174 | 0.299 | 1.998 | 0.912 |
| | $L_3$ | 0.398 | 0.407 | 0.319 | 0.351 | 0.468 | 0.457 | 0.415 | 0.427 | 0.214 | 0.305 | 0.274 | 0.293 | 0.319 | 0.408 | 1.979 | 0.912 |
| | $L_4$ | 0.454 | 0.440 | 0.420 | 0.410 | 0.469 | 0.477 | 0.425 | 0.444 | 0.254 | 0.335 | 0.351 | 0.343 | 0.875 | 0.701 | 2.109 | 0.938 |
| | Avg | 0.385 | 0.399 | 0.293 | 0.334 | 0.442 | 0.448 | 0.378 | 0.403 | 0.216 | 0.305 | 0.253 | 0.276 | 0.364 | 0.404 | 2.108 | 0.929 |
| w/o instructions | $L_1$ | 0.427 | 0.432 | 0.214 | 0.293 | 0.412 | 0.415 | 0.318 | 0.357 | 0.202 | 0.292 | 0.198 | 0.247 | 0.115 | 0.240 | 2.383 | 1.045 |
| | $L_2$ | 0.460 | 0.448 | 0.273 | 0.327 | 0.460 | 0.438 | 0.405 | 0.405 | 0.205 | 0.296 | 0.250 | 0.287 | 0.204 | 0.326 | 2.357 | 1.035 |
| | $L_3$ | 0.499 | 0.473 | 0.332 | 0.364 | 0.499 | 0.466 | 0.435 | 0.434 | 0.219 | 0.310 | 0.304 | 0.323 | 0.349 | 0.431 | 2.364 | 1.038 |
| | $L_4$ | 0.531 | 0.489 | 0.424 | 0.412 | 0.492 | 0.476 | 0.430 | 0.441 | 0.259 | 0.341 | 0.379 | 0.371 | 0.888 | 0.716 | 2.421 | 1.045 |
| | Avg | 0.479 | 0.461 | 0.311 | 0.349 | 0.466 | 0.449 | 0.397 | 0.409 | 0.221 | 0.310 | 0.283 | 0.307 | 0.389 | 0.428 | 2.381 | 1.041 |
| w/o mask | $L_1$ | 0.334 | 0.376 | 0.180 | 0.265 | 0.420 | 0.434 | 0.298 | 0.350 | 0.191 | 0.280 | 0.178 | 0.220 | 0.096 | 0.219 | 2.636 | 1.129 |
| | $L_2$ | 0.369 | 0.394 | 0.245 | 0.306 | 0.455 | 0.451 | 0.379 | 0.397 | 0.194 | 0.284 | 0.222 | 0.258 | 0.191 | 0.312 | 2.646 | 1.136 |
| | $L_3$ | 0.398 | 0.413 | 0.308 | 0.348 | 0.484 | 0.469 | 0.414 | 0.429 | 0.208 | 0.298 | 0.276 | 0.297 | 0.336 | 0.422 | 2.611 | 1.104 |
| | $L_4$ | 0.460 | 0.448 | 0.411 | 0.407 | 0.478 | 0.489 | 0.429 | 0.448 | 0.248 | 0.330 | 0.352 | 0.345 | 0.893 | 0.715 | 2.531 | 1.076 |
| | Avg | 0.390 | 0.408 | 0.286 | 0.332 | 0.459 | 0.461 | 0.380 | 0.406 | 0.210 | 0.298 | 0.257 | 0.280 | 0.379 | 0.417 | 2.606 | 1.112 |
| w/o LightTrans | $L_1$ | 0.333 | 0.370 | 0.185 | 0.269 | 0.393 | 0.414 | 0.301 | 0.348 | 0.204 | 0.292 | 0.182 | 0.225 | 0.094 | 0.218 | 2.549 | 1.045 |
| | $L_2$ | 0.371 | 0.388 | 0.249 | 0.311 | 0.437 | 0.440 | 0.379 | 0.396 | 0.206 | 0.295 | 0.226 | 0.262 | 0.186 | 0.308 | 2.365 | 0.998 |
| | $L_3$ | 0.403 | 0.409 | 0.321 | 0.353 | 0.471 | 0.456 | 0.419 | 0.428 | 0.220 | 0.308 | 0.280 | 0.300 | 0.334 | 0.420 | 2.157 | 0.971 |
| | $L_4$ | 0.460 | 0.441 | 0.425 | 0.412 | 0.471 | 0.470 | 0.430 | 0.446 | 0.259 | 0.338 | 0.356 | 0.349 | 0.887 | 0.711 | 2.141 | 0.978 |
| | Avg | 0.392 | 0.402 | 0.295 | 0.336 | 0.443 | 0.445 | 0.382 | 0.405 | 0.222 | 0.308 | 0.261 | 0.284 | 0.375 | 0.414 | 2.303 | 0.998 |
| w/o reconstruction | $L_1$ | 0.331 | 0.370 | 0.183 | 0.266 | 0.394 | 0.416 | 0.300 | 0.350 | 0.202 | 0.293 | 0.178 | 0.220 | 0.093 | 0.215 | 2.330 | 0.968 |
| | $L_2$ | 0.374 | 0.393 | 0.250 | 0.312 | 0.431 | 0.437 | 0.381 | 0.400 | 0.204 | 0.299 | 0.224 | 0.259 | 0.189 | 0.309 | 2.174 | 0.955 |
| | $L_3$ | 0.403 | 0.413 | 0.320 | 0.353 | 0.465 | 0.457 | 0.418 | 0.430 | 0.218 | 0.312 | 0.279 | 0.298 | 0.339 | 0.422 | 2.112 | 0.943 |
| | $L_4$ | 0.460 | 0.445 | 0.423 | 0.413 | 0.465 | 0.476 | 0.431 | 0.447 | 0.257 | 0.342 | 0.355 | 0.347 | 0.911 | 0.720 | 2.171 | 0.956 |
| | Avg | 0.392 | 0.405 | 0.294 | 0.336 | 0.439 | 0.447 | 0.383 | 0.407 | 0.220 | 0.312 | 0.259 | 0.281 | 0.383 | 0.417 | 2.197 | 0.956 |
| w/o all | $L_1$ | 0.445 | 0.436 | 0.218 | 0.297 | 0.425 | 0.438 | 0.308 | 0.350 | 0.199 | 0.290 | 0.193 | 0.237 | 0.116 | 0.241 | 2.223 | 1.027 |
| | $L_2$ | 0.473 | 0.451 | 0.277 | 0.332 | 0.457 | 0.454 | 0.395 | 0.399 | 0.201 | 0.294 | 0.243 | 0.276 | 0.206 | 0.326 | 2.522 | 1.095 |
| | $L_3$ | 0.498 | 0.470 | 0.329 | 0.361 | 0.496 | 0.465 | 0.423 | 0.428 | 0.217 | 0.307 | 0.296 | 0.313 | 0.354 | 0.433 | 2.569 | 1.104 |
| | $L_4$ | 0.533 | 0.490 | 0.426 | 0.417 | 0.499 | 0.479 | 0.436 | 0.449 | 0.258 | 0.339 | 0.370 | 0.361 | 0.902 | 0.721 | 2.601 | 1.110 |
| | Avg | 0.487 | 0.462 | 0.313 | 0.352 | 0.469 | 0.459 | 0.391 | 0.407 | 0.219 | 0.308 | 0.276 | 0.297 | 0.395 | 0.430 | 2.479 | 1.084 |
| TS-Text | $L_1$ | 0.330 | 0.368 | 0.185 | 0.269 | 0.392 | 0.416 | 0.299 | 0.349 | 0.199 | 0.291 | 0.182 | 0.224 | 0.095 | 0.215 | 2.276 | 1.015 |
| | $L_2$ | 0.372 | 0.389 | 0.250 | 0.309 | 0.435 | 0.439 | 0.375 | 0.396 | 0.203 | 0.296 | 0.226 | 0.262 | 0.186 | 0.307 | 2.232 | 1.020 |
| | $L_3$ | 0.404 | 0.410 | 0.322 | 0.355 | 0.474 | 0.466 | 0.418 | 0.428 | 0.218 | 0.310 | 0.281 | 0.300 | 0.332 | 0.418 | 2.247 | 1.019 |
| | $L_4$ | 0.459 | 0.445 | 0.424 | 0.413 | 0.481 | 0.485 | 0.431 | 0.449 | 0.258 | 0.340 | 0.355 | 0.348 | 0.910 | 0.717 | 2.275 | 1.016 |
| | Avg | 0.391 | 0.403 | 0.295 | 0.337 | 0.446 | 0.452 | 0.381 | 0.406 | 0.220 | 0.309 | 0.261 | 0.284 | 0.381 | 0.414 | 2.258 | 1.018 |
| Random Init | $L_1$ | 0.344 | 0.378 | 0.183 | 0.268 | 0.400 | 0.419 | 0.297 | 0.348 | 0.201 | 0.291 | 0.180 | 0.221 | 0.093 | 0.216 | 2.404 | 1.065 |
| | $L_2$ | 0.386 | 0.398 | 0.250 | 0.309 | 0.439 | 0.443 | 0.377 | 0.395 | 0.203 | 0.295 | 0.225 | 0.259 | 0.184 | 0.306 | 2.323 | 1.038 |
| | $L_3$ | 0.415 | 0.418 | 0.327 | 0.359 | 0.471 | 0.460 | 0.415 | 0.428 | 0.218 | 0.309 | 0.279 | 0.298 | 0.330 | 0.416 | 2.302 | 1.028 |
| | $L_4$ | 0.470 | 0.449 | 0.426 | 0.418 | 0.473 | 0.482 | 0.426 | 0.444 | 0.258 | 0.339 | 0.354 | 0.347 | 0.890 | 0.713 | 2.313 | 1.041 |
| | Avg | 0.404 | 0.411 | 0.297 | 0.339 | 0.446 | 0.451 | 0.379 | 0.404 | 0.220 | 0.309 | 0.260 | 0.281 | 0.374 | 0.413 | 2.336 | 1.043 |
| Freeze PLM | $L_1$ | 0.340 | 0.378 | 0.185 | 0.269 | 0.399 | 0.422 | 0.296 | 0.347 | 0.203 | 0.297 | 0.182 | 0.226 | 0.090 | 0.210 | 2.657 | 1.121 |
| | $L_2$ | 0.378 | 0.397 | 0.253 | 0.313 | 0.436 | 0.444 | 0.376 | 0.397 | 0.207 | 0.301 | 0.228 | 0.262 | 0.181 | 0.302 | 2.473 | 1.078 |
| | $L_3$ | 0.409 | 0.417 | 0.324 | 0.356 | 0.470 | 0.462 | 0.414 | 0.430 | 0.222 | 0.315 | 0.281 | 0.299 | 0.323 | 0.412 | 2.416 | 1.054 |
| | $L_4$ | 0.465 | 0.447 | 0.425 | 0.415 | 0.470 | 0.479 | 0.427 | 0.447 | 0.263 | 0.344 | 0.355 | 0.346 | 0.899 | 0.713 | 2.376 | 1.057 |
| | Avg | 0.398 | 0.410 | 0.297 | 0.338 | 0.444 | 0.452 | 0.378 | 0.405 | 0.224 | 0.314 | 0.262 | 0.283 | 0.373 | 0.409 | 2.481 | 1.078 |
| FPT PLM | $L_1$ | 0.332 | 0.373 | 0.182 | 0.265 | 0.395 | 0.416 | 0.297 | 0.346 | 0.199 | 0.291 | 0.180 | 0.225 | 0.091 | 0.212 | 2.316 | 1.042 |
| | $L_2$ | 0.371 | 0.393 | 0.255 | 0.311 | 0.433 | 0.439 | 0.375 | 0.395 | 0.202 | 0.294 | 0.226 | 0.262 | 0.183 | 0.304 | 2.243 | 1.024 |
| | $L_3$ | 0.403 | 0.415 | 0.321 | 0.352 | 0.464 | 0.456 | 0.414 | 0.429 | 0.219 | 0.313 | 0.279 | 0.298 | 0.331 | 0.416 | 2.287 | 1.023 |
| | $L_4$ | 0.459 | 0.445 | 0.423 | 0.414 | 0.458 | 0.472 | 0.426 | 0.443 | 0.259 | 0.343 | 0.354 | 0.346 | 0.898 | 0.716 | 2.297 | 1.023 |
| | Avg | 0.391 | 0.407 | 0.295 | 0.336 | 0.438 | 0.446 | 0.378 | 0.403 | 0.220 | 0.310 | 0.260 | 0.283 | 0.376 | 0.412 | 2.286 | 1.028 |

ratio compared to a smaller one. The best performance is generally obtained when the ratio is set to 0.5. As for the number of layers in the Language-TS Transformer, a count of 6 appears to be the most favorable choice. We refrain from setting the number of layers to a larger value, such as 7, due to constraints imposed by GPU memory limitations.

