# OpenReview forum: "UniTime: A Language-Empowered Unified Model for Cross-Domain Time Series Forecasting"
_ACM.org/TheWebConf/2024/Conference — TheWebConf24_

### Official Review · Reviewer_iGCs · 2023-11-23

**Novelty:** 5
**Technical Quality:** 5

**Review:**

The paper proposes UniTime for effective cross-domain time series learning.

Advantage:
1. The topic is interesting and important.
2. The paper proposes novel methods to solve cross-domain time series learning problems.
3. In experiments, the proposed method achieves better performance compared to baselines.

Disadvantage:
1. The motivation should be explained more. From my perspective, the task is not a generation task, and GPT2 may be not the best choice. Did you try T5 architecture?
2. Based on the ablation study, it seems that masking does not provide a huge performance improvement. Would you like to explain more and provide more insights?

**Questions:**

Please refer to the disadvantages.

**Reviewer Confidence:**

3: The reviewer is confident but not certain that the evaluation is correct

**Scope:**

4: The work is relevant to the Web and to the track, and is of broad interest to the community

---

### Official Review · Reviewer_AtQ9 · 2023-11-25

**Novelty:** 5
**Technical Quality:** 5

**Review:**

The paper introduces UniTime, a novel approach to multivariate time series forecasting, advocating for a unified model paradigm that transcends domain boundaries. The research addresses key challenges in cross-domain modeling, such as disparities in data characteristics and the difficulties in distinguishing data from various domains​​. UniTime is an innovative solution for learning from cross-domain time series data, employing human-crafted instructions for domain identification, and a Language-TS Transformer for aligning different input spaces to a common latent space of language models. This approach allows for flexibility in handling varying data characteristics and mitigates domain convergence speed imbalance​​.

The experimental design is comprehensive and the proposed method shows consistent performance improvement over state-of-the-art baselines. The ablation studies help identify the role of each component.

**Questions:**

Could you elaborate on why other baseline models, unlike UniTime, GPT4TS, and PatchTST, are not trained across multiple datasets? What specific challenges or limitations prevent these models from being effectively applied in a cross-dataset training context as done with UniTime?

**Reviewer Confidence:**

2: The reviewer is willing to defend the evaluation, but it is likely that the reviewer did not understand parts of the paper

**Scope:**

3: The work is somewhat relevant to the Web and to the track, and is of narrow interest to a sub-community

---

### Official Review · Reviewer_pxVF · 2023-11-26

**Novelty:** 3
**Technical Quality:** 4

**Review:**

Pros:

1. The paper is well-structured and easy to comprehend, offering a clear presentation of its contents.

2. The technique proposed in the study is logically sound and well-founded.

Cons:

1. The study's relevance to web mining is questionable. The method primarily addresses challenges common to general MTS tasks, without specifically focusing on unique issues associated with web-related scenarios.

2. The paper potentially overstates its contribution. It is not the first to explore using a unified model for generalization across time series application domains, as evidenced by:
[1] Tian Zhou, Peisong Niu, Xue Wang, Liang Sun, and Rong Jin's 2023 work, "One Fits All: Power General Time Series Analysis by Pretrained LM," published in Advances in Neural Information Processing Systems.
[2] Gruver N, Finzi M A, Qiu S, et al.'s 2023 paper, "Large Language Models Are Zero-Shot Time Series Forecasters," presented at the Thirty-seventh Conference on Neural Information Processing Systems.

3. The applicability of GPT to time series data is a matter of debate, considering the significant differences in data modality and distribution between text and time series data. The authors should provide a more thorough explanation and analysis to clarify the similarities between text data and time series data.

4. The paper's argument for the benefits of transferring knowledge across different time series datasets remains under-explored. Incorporating some quantitative analysis and results could strengthen this claim and make it more persuasive.

**Questions:**

Please address the aforementioned concerns.

**Reviewer Confidence:**

4: The reviewer is certain that the evaluation is correct and very familiar with the relevant literature

**Scope:**

1: The work is irrelevant to the Web

---

### Official Review · Reviewer_QZrP · 2023-11-26

**Novelty:** 5
**Technical Quality:** 4

**Review:**

This study utilizes language models to perform cross-domain time series forecasting with a unified model. Challenges in cross-domain time series forecasting, such as differences in data characteristics between domains and domain confusion Issues, are addressed by leveraging the knowledge from the language model. Information for each domain is provided to the model in the form of language, enabling the model to utilize information for the target domain. Through the use of language, mutually beneficial information between domains is appropriately utilized, and domain-specific information is acquired. Experimental results illustrate the proposed model's superior performance in cross-domain scenarios.

Pros:
- The utilization of a language model for cross-domain time series forecasting in this study is particularly interesting.
- Experimental results demonstrate that the proposed approach, involving jointly learned models from various domains, outperforms models trained individually in each domain.

Cons:
- Ultimately, hand-crafted domain instructions are required for each domain. This approach relies on the labor of individuals with domain-specific knowledge.
- In addition to the above, the incorporation of such domain instructions, in contrast to the baseline, makes it challenging to claim strict fairness in comparisons with other baselines. Nevertheless, the effectiveness of this approach is significant.

**Questions:**

Please see the above review.

**Reviewer Confidence:**

3: The reviewer is confident but not certain that the evaluation is correct

**Scope:**

3: The work is somewhat relevant to the Web and to the track, and is of narrow interest to a sub-community

---

### Official Review · Reviewer_bumz · 2023-11-29

**Novelty:** 5
**Technical Quality:** 5

**Review:**

The paper titled "UniTime: A Language-Empowered Unified Model for Cross-Domain Time Series Forecasting" introduces an innovative approach to time series forecasting, addressing the challenges of handling data across various domains. The paper identifies key issues in cross-domain time series forecasting, such as varying data characteristics, domain confusion, and domain convergence speed imbalance, and proposes the UniTime model as a solution. The key solution proposed by the authors is to incorporate natural language as the mediator across domains and help train a single model for multiple domains.

UniTime is a novel model that integrates domain instructions and a Language-TS Transformer to address these challenges. It is designed to be flexible and adaptable to data with varying characteristics, employing domain instructions to provide explicit identification information and align two modalities (text and time series). The model also uses masking to mitigate domain convergence speed imbalance issues.

Experiments demonstrate the effectiveness of UniTime, showing advancements in forecasting performance and zero-shot transferability. The model outperforms state-of-the-art methods on multiple public benchmark datasets. However, the paper has noticeable limitations, such as the complexity of the model and its implementation, the dependency on the quality of domain instructions, and potential scalability issues in real-world applications.

**Questions:**

1) I believe this method is likely to require large-scale finetuning of the language encoder model with domain-specific data - i.e., text data that provides descriptions of time series behaviors across different applications/application areas. How would the authors open source such a dataset? It might prove somewhat challenging to acquire.

2) In the absence of the finetuning step described in step 1, how do we ensure homogeneity of the instructions? What if some instructions are very short and others have a longer more-detailed explanation? I find it unlikely that the language representations would handle such corner cases unless the model is explicitly trained to handle such descriptions, and/or the format for the instructions is standardized so that every domain description carries the same structure and/or similar language.

3) While the current performance is impressive, I would like the authors to provide a more detailed study of the above two considerations to help readers better appreciate the weaknesses and strengths of relying on language-grounded generalization.

**Ethics Review Description:**

Not required.

**Reviewer Confidence:**

3: The reviewer is confident but not certain that the evaluation is correct

**Scope:**

3: The work is somewhat relevant to the Web and to the track, and is of narrow interest to a sub-community

---

### Decision · Program_Chairs · 2024-01-22

**Decision:**

Accept

**Comment:**

The research advocates for a unified model paradigm in multivariate time series forecasting, addressing challenges like domain disparities and convergence rates. The proposed UniTime model effectively adapts to varying data characteristics, employs domain instructions and a Language-TS Transformer for alignment, and uses masking to alleviate convergence speed imbalances, demonstrating improved forecasting performance in experiments.

 + Cross-domain time series forecasting is a interesting application of LLMs
 + Experimental results show a clear advantage of the proposed methodology
 - The hand-crafted prompts is a weakness in this approach as it limits the scale and relies on domain expertise for prompt creation.